# Nurturing through Nutrition: Exploring the Role of Antioxidants in Maternal Diet during Pregnancy to Mitigate Developmental Programming of Chronic Diseases

**DOI:** 10.3390/nu15214623

**Published:** 2023-10-31

**Authors:** Mariana S. Diniz, Carina C. Magalhães, Carolina Tocantins, Luís F. Grilo, José Teixeira, Susana P. Pereira

**Affiliations:** 1CNC-UC—Center for Neuroscience and Cell Biology, University of Coimbra, 3004-504 Coimbra, Portugal; mdiniz@cnc.uc.pt (M.S.D.); cmagalhaes@cnc.uc.pt (C.C.M.); ctsantos@cnc.uc.pt (C.T.); luis.grilo@uc.pt (L.F.G.); 2CIBB—Centre for Innovative Biomedicine and Biotechnology, University of Coimbra, 3004-517 Coimbra, Portugal; 3Doctoral Programme in Experimental Biology and Biomedicine (PDBEB), Institute for Interdisciplinary Research, University of Coimbra, 3004-504 Coimbra, Portugal; 4Laboratory of Metabolism and Exercise (LaMetEx), Research Centre in Physical Activity, Health and Leisure (CIAFEL), Laboratory for Integrative and Translational Research in Population Health (ITR), Faculty of Sports, University of Porto, 4200-450 Porto, Portugal

**Keywords:** maternal antioxidant supplementation, disease prevention, chronic diseases, developmental programming, metabolic dysfunction, oxidative stress

## Abstract

Chronic diseases represent one of the major causes of death worldwide. It has been suggested that pregnancy-related conditions, such as gestational diabetes mellitus (GDM), maternal obesity (MO), and intra-uterine growth restriction (IUGR) induce an adverse intrauterine environment, increasing the offspring’s predisposition to chronic diseases later in life. Research has suggested that mitochondrial function and oxidative stress may play a role in the developmental programming of chronic diseases. Having this in mind, in this review, we include evidence that mitochondrial dysfunction and oxidative stress are mechanisms by which GDM, MO, and IUGR program the offspring to chronic diseases. In this specific context, we explore the promising advantages of maternal antioxidant supplementation using compounds such as resveratrol, curcumin, *N*-acetylcysteine (NAC), and Mitoquinone (MitoQ) in addressing the metabolic dysfunction and oxidative stress associated with GDM, MO, and IUGR in fetoplacental and offspring metabolic health. This approach holds potential to mitigate developmental programming-related risk of chronic diseases, serving as a probable intervention for disease prevention.

## 1. Introduction

Non-communicable chronic diseases (CD) are considered to be one of the major threats to global health and include cardiovascular disease, diabetes, obesity, cancer, chronic respiratory diseases, and chronic liver disease [1]. It is estimated that, in 2016, non-communicable CD contributed to two-thirds of mortality worldwide [2]. There are several CD risks factors an individual can manage, including high blood pressure, tobacco smoking, high body mass index, physical inactivity, and constant consumption of poor diets [1]. In addition, unmanageable risk factors can also contribute to CD development, such as age, sex, and genetic background [3]. On top of that, recent research has suggested that the intrauterine environment, which is modulated by maternal behaviors and disease, including gestational diabetes mellitus (GDM) [4,5], maternal obesity (MO) [5,6,7,8], and IUGR [9], severely influences the offspring’s CD development risk.

Gestational diabetes mellitus (GDM) is defined as hyperglycemic and glucose intolerance states that are detected, for the first time, either in the second or third trimester of pregnancy [10]. It is well established that GDM is the most prevalent pregnancy complication, affecting 13.9% of pregnancies [4,11]. Identified risk factors for GDM development encompass maternal age and obesity, with obese pregnant women presenting a 2.4-fold higher risk of developing GDM [12]. Furthermore, MO itself represents a highly prevalent pregnancy complication [5]. It is estimated that approximately 50% of pregnancies occur in overweight or obese women [6]. Both GDM and MO contribute to an increased risk of inducing a fetoplacental environment resembling prolonged hypoxia, along with other factors, such as maternal smoking, vascular dysfunction, and maternal nutrient reduction [13]. These conditions can potentiate suboptimal fetal development and growth, a condition referred to as intra-uterine growth restriction (IUGR). Indeed, neonatal complications associated with GDM, MO, and IUGR include increased perinatal mortality and morbidity, deviations in birthweight, and preterm birth [14,15,16]. These pregnancy-associated disorders induce structural, functional, and metabolic adaptations across several organs as early as the fetal stage. In this context, it has been pointed out that oxidative stress and mitochondrial dysfunction may be pivotal mechanisms of developmental programming in pregnancy-related disorders [7]. Therefore, these mechanisms can be strategic targets to modulate the programming of non-communicable CD in offspring during pregnancy.

Despite inconclusive and controversial data, studies have explored the use of antioxidant supplementation for CD treatment. Recent research has highlighted the promising beneficial effects of maternal supplementation with natural and/ or synthetic antioxidants to mitigate the developmental programming of chronic diseases. Herein, we delve into the compelling body of evidence on possible mechanisms of offspring’s chronic disease programming by maternal health and discuss possible beneficial effects of supplementation with antioxidant compounds such as vitamins, resveratrol, curcumin, N-acetylcysteine (NAC), and Mitoquinone (MitoQ), which have garnered attention due to their beneficial potential explored in a context of developmental programming by maternal habits and in their ability to safeguard the long-term wellbeing of offspring.

## 2. Developmental Programming of Chronic Diseases by Pregnancy-Associated Disorders

### 2.1. (Patho)physiologic Role of Reactive Oxygen Species in Fetal Development

Proper fetal development hinges on an interplay of several critical factors, among which a consistent and uninterrupted provision of nutrients and oxygen plays a pivotal role [17]. Oxygen levels exhibit a fine orchestration throughout gestation according to the specific requirements of the developing fetus [18]. In the early stages, oxygen is maintained at lower levels, particularly up until the 12th week following conception [19]. This intentionally lowered oxygen levels stimulate angiogenesis, promoting the formation of new blood vessels, which is a vital process for sustaining early development [19]. Around the 16th week of gestation, a significant shift occurs, with intrauterine oxygen levels increasing significantly and then remaining stable until birth [19]. Even in this carefully regulated environment, a small fraction of the oxygen required for oxidative metabolism undergoes incomplete reduction, giving rise to reactive oxygen species (ROS) [19]. ROS production predominantly occurs via the escape of electrons from the mitochondrial electron transport chain (ETC), with complex-I and -III being notable contributors, resulting in the formation of the superoxide (•O_2_^−^) radical. Notwithstanding, other sources of ROS production can be considered, such as dihydroorotate dehydrogenase (DHODH), among others. It is important to state that ROS have a biphasic effect [20]. On the one hand, at moderate levels, ROS are key players in pregnancy physiology, acting as signaling molecules in developmental processes including placental growth [21], embryo development, and implantation [22], and are involved in the replication, differentiation, and maturation of cells and organs. However, on the flip side, excessive ROS levels, when not counterbalanced by the antioxidant capacity, can usher in oxidative stress [5]. This state of oxidative stress can inflict severe damage, compromising the structural integrity of cells and organelles’ membranes and hindering proper protein function. Furthermore, it poses a significant risk to fetal development and the intricate process of proper organogenesis.

In sum, the orchestration of oxygen levels and the balance of ROS levels within the maternal–fetal interface are pivotal to ensuring the proper progression of fetal development.

### 2.2. The Impact of Pregnancy-Associated Disorders on Offspring’s Organs Oxidative Stress and Mitochondrial Function

Pregnancy is a state of high energetic demand that is sustained mainly by fetoplacental metabolic activity [23]. Mitochondria are metabolism’s key players and one of the main cellular energy sources [24], being highly responsive organelles to energy demands. Consequently, placental mitochondrial function can be modulated [24] in response to an adverse intra-uterine environment. In pregnancy-related disorders, placental metabolic dysfunction has been extensively documented, which may translate into long-lasting consequences for the offspring in the prenatal and postnatal periods. This section aims to discuss how GDM, MO, and IUGR are related with fetoplacental dysfunction and offspring organ dysfunction via mitochondrial malfunction and oxidative stress.

The characteristic hyperglycemic state during GDM may adversely impact the placental mitochondrial structure and function [25] (Figure 1). Indeed, placentas from GDM-portraying women presented swollen and disrupted mitochondria, some of which were completely damaged [26]. Since mitochondria are highly dynamic organelles, changing the number, morphology, network, and size according to cellular energy needs, mitochondrial shape and function are tightly linked [27]. For instance, GDM-derived human cytotrophoblasts present a decreased mitochondrial maximum respiratory capacity and decreased ATP production rates [28,29], highlighting GDM-induced mitochondrial bioenergetic dysfunction (Figure 1). Maternal diabetes, either pregestational type 2 diabetes or GDM, impair placental mitochondrial biogenesis (formation of new mitochondria), with decreased mitochondrial transcription factor A (TFAM) [30] and peroxisome proliferator-activated receptor gamma coactivator 1-alpha (PGC1-α) expression levels [28,29]. In addition to impaired placental mitochondrial biogenesis, GDM leads to altered mitochondrial dynamics, with studies in humans reporting either an increase in placental mitochondrial fusion events [28,31] or a decrease [32]. Fusion events have been suggested as essential for mitochondrial DNA (mtDNA) copy number maintenance [33] (Figure 1). Although inconsistent, mtDNA copy number alterations are reported in GDM-related studies, either in maternal serum [34,35] or placental tissue [28,30]. A potential direct relationship between the mtDNA copy number and oxidative stress has been suggested [36]. This hypothesis was raised because the placental mtDNA copy number was positively correlated with placental DNA oxidation [36] both in GDM and control pregnancies. Further studies are required to understand the mechanisms linking DNA oxidation and the mtDNA copy number to solidify this hypothesis. Despite this, placental oxidative stress has been widely reported in GDM placentas. Although ROS are generated from several sources, mitochondria are considered as one of the major ones [5]. GDM human pregnancies present placental increased biomarkers of oxidative stress, such as malondialdehyde (MDA) [37,38], reduced antioxidant defenses (decreased catalase (CAT) activity [38], and glutathione peroxidase (GPx) 1 [39]) (Figure 1). In addition to the placenta, GDM human umbilical cord mesenchymal stem cells (hUC-MSC) present increased ROS production, detected via 2′,7′-Dichlorofluorescin diacetate (DCFDA), and impaired mitochondrial bioenergetics, including a diminished basal respiration state and FCCP-induced maximum respiratory capacity [40]. Given that mesenchymal stem cells (MSC) are multipotent, they can differentiate into a wide range of cell types during fetal development, including adipocytes, cardiomyocytes, myocytes, and neurons [41]. Dysfunctional MSC may imprint dysfunctional cells and organs in the offspring that may later lead to an increased CD risk. GDM impairment in the offspring’s organs remains underexplored in the literature, demanding increased and larger studies. Nonetheless, the available data, both in humans and in murine animal models, suggests that GDM can lead to mitochondrial-related alterations in offspring’s preadipocytes [42], pancreas [43], liver [44], and heart [45,46,47]. In addition to this, in rat embryos, maternal hyperglycemia has been associated with increased lipid peroxidation, indicated by increased MDA levels. In addition, animal rodent studies have shown the presence of oxidative stress in GDM offspring’s tissues, such as in the cerebral cortex marked by increased ROS (detected via H_2_-DCF-DA), increased lipid peroxidation, and decreased CAT activity in comparison with the respective control in both adult male and female rat offspring of streptozotocin-treated mothers [48]. The existing data on this topic is limited, underscoring the need for more comprehensive and in-depth research.

It has been suggested that the intrauterine metabolic milieu is different for GDM and MO [49] (Figure 1). Nonetheless, despite these differences between GDM and MO, MO’s intrauterine environment is also hyperglycemic and pro-oxidative. Similar to GDM, MO human placentas present mitochondrial dysfunction, i.e., disrupted biogenesis (evidenced by decreased PGC-1α protein expression levels and decreased citrate synthase activity) [50] and bioenergetics (decreased complex-I activity and decreased ATP levels) [50,51], excessive ROS formation and oxidative damage (MDA and protein carbonylation increased levels, increased DCF fluorescence) [50,51,52], and lower antioxidant defenses (decreased SOD activity, decreased GPx-4 expression levels) [50,53], either in human whole placental tissue or in cytotrophoblasts (Figure 1).

This modified metabolic environment has been suggested to contribute to MO-induced placental metabolic dysfunction, highlighting the potential impact of MO-induced placental mitochondrial dysfunction on fetal growth. Animal models have shown that MO induces mitochondrial dysfunction in the offspring’s organs postnatally, with the liver [54,55,56,57,58] being the most studied and reported in the literature in murine [55,56] and ovine animal models [57]. The majority describes decreased hepatic mitochondrial respiratory chain (MRC) complex activities [54,57] and expression [55], except for the work by Alfaradhi et al., of which offspring’s hepatic MRC complex activities were described to be increased [56]. MO-induced offspring’s hepatic MRC alterations are evident, at least in the early stages after birth. Only a limited number of these studies explored mitochondrial dynamics and biogenesis. Therefore, we consider it essential to investigate the pathways involved in these mechanisms, given their significant role in mitochondrial fitness [59]. Mitochondrial alterations in MO’s offspring were also described in other tissues, namely in murine [60,61,62,63,64,65,66] and non-human primate studies [67], reporting alterations in the offspring’s cardiac tissue [60,61,62,64], skeletal muscle [63,67], and hypothalamus [65]. Importantly, the majority of these studies consistently report impaired mitochondrial oxygen consumption across various time points in the offspring’s life. However, the increased fusion events were only reported for MO offspring’s hypothalamus. Furthermore, in both murine and non-human primate animal models, there is compelling evidence of diminished antioxidant defenses in MO offspring’s hearts [68], and livers [56,69,70] and these include decreased SOD2, GPx1 expression levels, and reduced-to-oxidized glutathione ratio (GSH/GSSG), as well as increased levels of oxidative stress markers (e.g., TBARS) in the pancreas [69], liver [69,70,71], and skeletal muscle [67].

Prolonged fetal exposure to MO- and GDM-induced hypoxia can lead to the development of IUGR. Notably, studies have identified commonalities between GDM, MO, and IUGR in the placenta, highlighting mitochondrial function impairment (i.e., increased mitochondrial content, decreased gene expression of MRC complexes’ subunits [72], and increased placental ATP content [73]) and oxidative stress (i.e., increased ROS measured using the fluorescence levels of H_2_-DCF-DA [74] and augmented catalase expression, possibly as a compensatory mechanism [75]) contributing to metabolic dysfunction in human IUGR placenta [72], specifically in a mice IUGR model achieved with maternal diet restriction [73] and in rat IUGR models achieved with hypoxia induction [74,75]. In IUGR offspring, mitochondrial dysfunction is verified across several organs, including the heart [76,77,78] and liver [79], and this is reported heterogeneously regarding the fetal or offspring’s age. Moreover, oxidative stress is also present in IUGR offspring’s brain and heart (increased lipid peroxidation biomarker MDA in rat IUGR model achieved with surgery [80] and in a non-human primate IUGR model achieved with maternal diet reduction [77]), as well as in the liver (increased lipid peroxidation biomarker 4-Hydroxynonenal (4-HNE) levels and decreased glutathione levels in a rat IUGR model achieved with diet restriction [81]) across different time-points in the fetal or offspring’s life.

In this context, the upcoming sections will delve into current research with antioxidants, particularly focusing on polyphenols, to gain a deeper understanding of the potential benefits of maternal supplementation with naturally occurring or modified antioxidants during pregnancy that might improve the overall offspring’s health, especially in cases of MO, GDM, and IUGR pregnancies.

## 3. Antioxidants and Mitochondriotropic Activity of Naturally Occurring and Synthetic Antioxidants

Developmental programming due to MO, GDM, and IUGR involves a complex interplay of tissue-specific and general mechanisms [82], in which epigenetic changes, mitochondrial changes, and oxidative stress are highlighted as the main players in this relationship [7]. Moreover, tissue and mitochondrial dysfunction commonly resulting from an exacerbation of pregnancy-associated oxidative stress results in maternal metabolic dysfunction in a positive feedback loop [83]. Thus, maternal supplementation with antioxidants represents a promising approach to mitigate developmental programming effects in the offspring exposed to pregnancy-related disorders characterized by elevated ROS. Antioxidant therapy has been studied as a possible strategy for exogenous antioxidants, such as those from dietary and synthetic sources, to act synergistically with endogenous antioxidants in restoring redox homeostasis [84].

### 3.1. Naturally Occurring Antioxidants

#### 3.1.1. Endogenous Antioxidants

Under physiological conditions, the human body naturally produces antioxidant molecules to eliminate excessive ROS, preventing oxidative stress and subsequent tissue damage [85]. The endogenous antioxidant defense system contains antioxidant enzymes such as SOD, CAT, and GPx, as well as non-enzymatic compounds such as glutathione, proteins like ferritin, and low molecular weight scavengers such as Coenzyme Q10 (CoQ10) that are responsible for maintaining cellular redox homeostasis [86]. SOD belongs to the first line of the antioxidant defense system by converting the superoxide anion radical to hydrogen peroxide (H_2_O_2_) [85,86]. In turn, both CAT and GPx are involved in the reduction of H_2_O_2_ to water and molecular oxygen [85,86]. In fact, enzymatic antioxidants are more effective in counteracting oxidative stress due to their ability to eliminate ROS, preventing damage to proteins, DNA, and lipids [85]. On the other hand, non-enzymatic antioxidants can act by capturing transition metal ions that are mostly responsible for producing reactive oxygen radical species or can store metal ions necessary for the synthesis of enzymes containing metal ions (metal-binding proteins, e.g., ferritin and transferrin) [85,86]. Moreover, they can scavenge reactive radicals (e.g., glutathione and uric acid) and directly interfere with the initiation and propagation steps of the peroxidation process, acting as inhibitors of lipid peroxidation [85,86,87] and protecting cells from harm, playing an important role in metabolism as is the case of CoQ10, a naturally occurring endogenous lipid-soluble antioxidant located in the inner mitochondrial membrane.

#### 3.1.2. Dietary Antioxidants

Dietary antioxidants such as Vitamin C and E, carotenoids, and polyphenols are naturally occurring exogenous antioxidants that complement the activity of the endogenous antioxidant defense system [86]. Research has been dedicated to unraveling the mechanisms underlying the actions of dietary phytochemicals, with a particular emphasis on polyphenols, and how polyphenols exert their beneficial effects, including their antioxidant capacity.

Polyphenols are subdivided into two major classes: flavonoids, which include quercetin and epigallocatechin-3-gallate (EGCG), and non-flavonoids, which include resveratrol and curcumin [88]. Dietary polyphenols present multiple benefits to human health, such as anti-cancer [89], antioxidant [90,91], anti-inflammatory [91,92], anti-obesity [91], and anti-diabetic [93]. Among the natural polyphenols, the most extensively researched compounds are resveratrol (3,4′,5-Trihydroxystilbene), curcumin (1E,6E)-1,7-bis(4-hydroxy-3-methoxyphenyl)hepta-1,6-diene-3,5-dione), EGCG, and quercetin, owing to their impact in biological properties.

Polyphenols have been considered a potential therapeutic approach due to their ability to mitigate oxidative stress events via ROS scavenging, modulating ROS-removing enzymes, inhibiting ROS-producing enzymes, or via the chelation of metals that are involved in metal-dependent hydroxyl formation via the Fenton reaction [88]. Although polyphenols have the capacity to reduce ROS, which is intrinsically related to their chemical structure and the presence of at least one phenol group, the indirect action of dietary polyphenols relies on the upregulation of endogenous antioxidant proteins [85]. It has been suggested that resveratrol activates SIRT1 by directly inhibiting phosphodiesterase (PDE) enzymes, increasing cAMP levels and AMPK activation with a subsequent increase in NAD+ levels activating SIRT1 [94]. SIRT1 activation is associated with a multitude of beneficial effects, namely the ability to decrease oxidative stress, inflammation, and regulation of the expression of mitochondrial genes involved in biogenesis and lipid metabolism (SIRT1/PGC-1α axis) [95,96,97,98,99,100,101,102,103,104]. Although the mechanism is not yet well known for curcumin’s action on SIRT1, this protein is also suggested to represent a target of curcumin. Additionally, research has suggested that curcumin’s antioxidant action also relies on its ROS scavenging properties, which are attributable to the compound’s structure. On top of the mechanisms of action described above, an important indirect mechanism to mitigate cellular oxidative stress is the activation of the nuclear factor erythroid 2–related factor 2 (Nrf2)/antioxidant responsive element (ARE) pathway. Both resveratrol [105] and curcumin [106,107,108] can induce the transcriptional activation of Nrf2 and subsequent upregulation of the expression of Nrf2 target genes, including NAD(P)H:quinone oxidoreductase 1 (NQO1) and heme oxygenase-1 (HO-1), conferring an additional antioxidant ability by increasing the expression of antioxidant enzymes.

In addition to the mitigation of oxidative stress, antioxidant compounds present effects in other biological processes, such as mitochondrial function and inflammation. For instance, resveratrol has been reported to induce BNIP3-related mitophagy and attenuate hyperlipidemia-related endothelial dysfunction [109] via the involvement of AMPK and hypoxia-inducible factor 1 (HIF-1). On the other hand, resveratrol effects can also contribute to increasing mitochondrial activity by regulating the expression of genes associated with oxidative phosphorylation, biogenesis, and lipid metabolism [95,97]. Curcumin’s main described mechanisms of action in mitochondria are targeting mitochondrial biogenesis, intrinsic apoptosis, mitochondrial permeability transition pore, mitochondrial MRC uncoupling, and ATP synthase [88]. Resveratrol and curcumin have also been associated with potential anti-inflammatory mechanisms. These compounds have demonstrated the ability to inhibit the nuclear factor-kB (NF-kB) signaling pathway, which subsequently prevents the expression of inflammatory cytokines such as tumor necrosis factor-alpha (TNF-α) and interleukin-6 (IL-6) (resveratrol [110,111], curcumin [112]).

### 3.2. Synthetic Antioxidants

In addition to naturally occurring antioxidants, synthetic antioxidant molecules such as N-Acetylcysteine (NAC), MitoTEMPO, and Mitoquinone (MitoQ) possess the capability to exert direct and indirect antioxidant effects in vivo [84,113,114]. These compounds have been developed to overcome mainly pharmacological limitations. For instance, NAC, a cysteine derivative, overcomes the cysteine low intracellular concentration limits, contributing to a more efficient rate of GSH synthesis [115,116]. Therefore, NAC not only makes a significant antioxidant contribution by serving as a precursor to intracellular GSH [116,117], but also directly interacts with certain free radicals and activates the Nrf2 signaling pathway, providing an effective means to combat oxidative stress at the cellular level [118].

In turn, as mitochondrial oxidative stress is associated with multiple diseases, targeting specifically mitochondrial ROS and modulating redox signaling may represent a more efficient antioxidant therapy [119]. However, it is still a challenge to specifically target and accumulate antioxidant molecules within the mitochondrial matrix [119]. MitoTEMPO and MitoQ are some examples of successful mitochondria-targeted antioxidants [119].

#### Mitochondria-Targeted Antioxidants

Although targeting mitochondria is not easily achievable, the linkage of an antioxidant molecule to a lipophilic cation, such as the triphenylphosphonium (TPP^+^) moiety, has been used to direct these molecules to the negatively charged mitochondrial matrix [114,120]. MitoQ, the mitochondria-targeted golden-standard antioxidant, is composed of a CoQ10 linked to a TPP^+^ moiety via a 10-carbon chain [121]. MitoQ’s remarkable antioxidant and antiapoptotic properties are primarily attributed to its ability to induce mitophagy, a selective process of degrading or removing damaged or dysfunctional mitochondria within a cell [122,123]. In addition to preventing ROS overproduction, MitoQ was also shown to decrease mitochondrial oxidative stress in in vitro and in vivo models of diabetic kidney disease by promoting the restoration of mitophagy via the Nrf2/PTEN-induced kinase (PINK) pathway [124,125]. Other molecules have been synthesized since then, such as MitoVitE, MitoTEMPOL, and SkQ1 [114]. In particular, mitochondria-targeted antioxidants based on caffeic acid (AntiOxCIN4) and gallic acid (AntiOxBEN2) were developed to overcome their highly hydrophilic character that makes it difficult to cross biological membranes [120] and target mitochondrial oxidative stress [120,126,127,128]. In fact, AntiOxCIN4 supplementation prevented hepatic steatosis in a non-alcoholic fatty liver disease (NAFLD) mice model [129], highlighting the potential of mitochondria-targeted therapeutic approaches for the treatment of non-communicable CDs.

Despite the wide variety of antioxidants being currently studied for the mitigation and/or prevention of non-communicable CD, maternal antioxidant supplementation studies are limited only to a few of these compounds, such as MitoQ, NAC, resveratrol, and curcumin. For this reason, in the next section, we explore the potential benefits of maternal supplementation with naturally occurring or modified antioxidants before and/or during pregnancy that might improve the overall offspring’s health, especially in cases of MO, GDM, and IUGR pregnancies.

## 4. Exploring the Role of Dietary Antioxidant Supplementation during Pregnancy in Offspring Non-Communicable Disease Prevention

Maternal nutritional status is of paramount relevance during the pre- and post-conception period and throughout lactation for optimal fetal development [130]. Evidence suggests that maternal dietary adjustments, such as maintaining a balanced energy and protein intake, can be beneficial during these phases to reduce the risk of fetal loss, stillbirth, and perinatal death [131]. Although a maternal high-protein diet has been associated with a higher percentage of small for gestational age (SGA) infants [131], the supplementation of specific amino acids has shown positive outcomes. Specifically, oral L-Arginine supplementation during pregnancy has been shown to improve fetoplacental circulation and birth weight in humans [132] and increase fetal viability and birth weight in pigs and sheep [132]. In a non-pregnant individual’s liver, it has been reported that L-Arginine can induce antioxidant response via stimulation of GSH synthesis and activation of the Nrf2 pathway, highlighting the potential role of L-Arginine to modulate antioxidant defenses [133]. Moreover, antenatal iron and folic acid supplementation are recommended by the WHO to prevent fetal and neonatal loss, SGA, maternal anemia, and iron deficiency [134].

In non-pregnant individuals with iron deficiency, the supplementation led to a significant decrease in oxidative stress [135], although it remains controversial regarding maternal supplementation, particularly in already-iron-sufficient mothers [136]. The overall diet’s antioxidant impact may vary, our main focus in this review is centered on specific antioxidant supplementation in pregnancy-associated disorders and its potential to mitigate offspring’s development of CDs. In this section, we discuss the literature concerning the impact of vitamins, resveratrol, curcumin, NAC, and MitoQ supplementation in pregnancies complicated by MO, GDM, and IUGR, exploring the consequent effects in the mother, in the fetoplacental unit, and in the offspring.

### 4.1. Vitamins

The debate on vitamin supplementation during pregnancy is extensive. Current research suggests that vitamin C and E supplementation during pregnancy may not reduce the risk of fetal or neonatal loss, SGA infants, nor preterm birth, with no observable beneficial effects [137,138,139]. Vitamin A supplementation during pregnancy has demonstrated positive outcomes in specific cases, such as improved fetal growth, in mothers experiencing night blindness, and those with HIV [140]. Additionally, supplementation in the postpartum period, when maternal vitamin A concentrations are low, has increased breast milk’s retinol content [140], potentially leading to reinforced immunity in these children. However, studies have shown that for some specific cases, high doses of, for example, vitamin E and beta-carotene, are associated with an increased risk of preterm birth. Thus, the safety and efficacy of antioxidant supplementation should be carefully assessed to reach optimal types, dosages, and timing of antioxidant supplementation for different populations of pregnant women. In light of these findings and taking into consideration that clinical trials involving pregnant women are still a subject of controversy [141], there is an emerging need to explore novel, safe, effective, and beneficial options for supplementation during pregnancy.

### 4.2. Resveratrol

In a pregnant obese rat model induced via a high-fat diet (HFD), maternal resveratrol supplementation decreased the number of large adipocytes and increased the number of small adipocytes in the mothers’ adipose tissue, contributing to the shift from hypertrophic to hyperplastic expansion of white adipose tissue (WAT) [142]. Maternal resveratrol supplementation also induced thermogenesis in MO offspring’s brown and white adipose tissues with increased thermogenic genes’ expression levels (e.g., PR/SET Domain 16 (PRDM16) and uncoupling protein 1 (UCP-1)) [142]. The mechanism of action may likely be mediated via the AMPK/SIRT1 pathway. This is supported by studies demonstrating that resveratrol promotes AMPK and SIRT1 phosphorylation, increasing their activity and downstream signaling [142,143].

In addition, oral supplementation of resveratrol in HFD-induced obese mothers decreased the expression of hepatic genes encoding for proteins involved both in glycolysis (i.e., Glucose-6-phosphate dehydrogenase (G-6-PDH), Phosphoenolpyruvate carboxykinase (PEPCK)) and in inflammation (i.e., IL-6), where it decreased the concentration of a DNA oxidation marker (8-Oxo-2′-deoxyguanosine—8-oxo-dG) and decreased the nitrotyrosine immunostained area in comparison with the MO group. These results suggest that resveratrol attenuated insulin resistance mechanisms, inflammatory processes, and oxidative stress in the maternal liver [144]. At 19 days of gestation, in the placental tissue, resveratrol supplementation in obese mothers carrying male fetuses decreased the levels of the lipid peroxidation biomarker (MDA) in comparison with the MO group, while for female fetuses, ROS levels were decreased along with SOD increased activity [144]. In the hepatic tissue of male offspring, maternal resveratrol supplementation increased GPx activity, while female hepatic tissue presented decreased DNA oxidation marker 8-oxo-dG, highlighting the antioxidant effect of resveratrol in a sex-specific way [144]. Hepatic Nrf2 activation may be involved in the improvement of antioxidant capacity induced via resveratrol [145].

The promising results observed with antioxidant supplementation in the context of type 2 diabetes [96,146] have prompted researchers to investigate its effects in a GDM context [147,148,149]. Notably, improvements in maternal glucose and insulin tolerance were reported in a GDM genetic mouse model (C57BL/KsJdb/+ (db/+)) treated with resveratrol via AMPK activation [147]. In addition, and as reported for mice GDM mothers’ liver, a study showed that resveratrol administration via oral gavage induces decreased hepatic glucose-6-phosphatase (G6Pase) activity in mice GDM offspring, contributing to the downregulation of the gluconeogenesis pathway [147].

### 4.3. Curcumin

Similar to resveratrol, curcumin administration via orogastric gavage in mothers on an HFD demonstrated the capacity to enhance insulin sensitivity and activate thermogenesis in brown adipose tissue (BAT) and WAT in male mice offspring (Figure 2) [150]. This effect is achieved by increasing the expression of thermogenic genes, such as PRDM16 and UCP-1 [150]. These favorable outcomes are likely to contribute to the improvement of mitochondrial function and energy expenditure in offspring born to mothers following an HFD [142], which have been suggested to be impaired by MO [66].

Notably, curcumin oral gavage administration also ameliorates GDM symptoms (e.g., hyperglycemia and insulin resistance) by promoting maternal hepatic AMPK activation and increasing the expression of GSH, SOD, and CAT in the livers of the GDM genetic mouse model (C57BL/KsJdb/+ (db/+)) [148]. In this study, it was suggested that AMPK upregulation may contribute to the inhibition of histone deacetylase 4 (HDAC4) and the subsequent downregulation of G6Pase protein expression and activity [148]. Thus, the effects of curcumin-induced AMPK increased activation on hepatic mitochondrial function and ROS production demand further investigation.

Maternal malnutrition has been associated with placental insufficiency and, consequently, with IUGR. Some studies reported that oral gavage administration and oral supplementation of curcumin induced beneficial effects on placental inflammation and oxidative damage via the regulation of the NF-kB and Nrf2 signaling pathways, respectively, in mouse and rat low protein diet-induced IUGR models [151,152], improving placental efficiency, alleviating placental apoptosis, and contributing to the loss of blood sinusoids area in the placental labyrinth [151,152].

Dietary polyphenol supplementation, such as resveratrol and curcumin, has already been demonstrated to prevent IUGR-induced inflammation and oxidative damage in the liver of the offspring of the IUGR spontaneous piglet model and IUGR rat model induced via bilateral artery ligation and protein restriction [153,154,155].

### 4.4. NAC

In a mice model of MO induced via an obesogenic diet, maternal NAC oral supplementation decreased WAT, hepatic fat, and inflammation and increased thermogenic gene expression in BAT [156]. Furthermore, an increase in GPx activity and rescue of the expression of some NADPH oxidase (NOX) enzyme complex genes to control the levels in the left ventricular cardiac tissue of male and female offspring were observed 7 days after birth, emphasizing the potential antioxidant effect of maternal NAC supplementation in the offspring’s postnatal health [157]. Additionally, maternal NAC supplementation has been shown to prevent increased MO-induced activation of the cardiac Akt-mTOR signaling pathway [157], which has been associated with the development of offspring’s cardiac hypertrophy [158]. However, MO offspring born to NAC-treated mothers developed cardiac hypertrophy [157], suggesting that another mechanism can be behind cardiac hypertrophy development. Nonetheless, in male offspring, maternal NAC supplementation improved MO-induced cardiac physiological abnormalities, including increased heart weight, interventricular septal thickness, and collagen content, potentially via the attenuation of oxidative stress, due to the role of ROS in the signaling of these pathways [157]. Although some evidence exists regarding the offspring’s cardiac benefits of antioxidant supplementation in obese mothers, further research is required to understand the mechanisms underlying these effects.

Oral gavage administration of NAC seems to decrease maternal oxidative stress by upregulating antioxidant defenses in the liver, including SOD, GPx, and CAT activities and GSH levels, and by reducing MDA levels in a GDM genetic animal mice model [149]. In addition, NAC contributed to redox homeostasis via the activation of Nrf2/HO-1 and decreased the expression of pro-inflammatory cytokines IL-6 and TNF-α in the maternal liver of a GDM mice genetic animal model [149]. Despite the relevant positive outcomes of antioxidant supplementation in GDM-portraying mothers, there are fewer results reported for the offspring. However, resveratrol [147], curcumin [148], and NAC [149] oral gavage administration have been demonstrated to contribute to the restoration in litter size and body weight at birth.

Interestingly, maternal supplementation with NAC in the drinking water of a IUGR model induced by implanting ameroid constrictors on uterine arteries of pregnant guinea pigs at mid-gestation was also demonstrated to play an important role in restoring fetal weight and placental efficiency by preventing endothelial dysfunction associated with IUGR [159]. NAC was shown to be able to revert the functional and epigenetic programming of endothelial nitric oxide synthase (eNOS) by reverting the reduced DNA methylation in the promoter region of the nitric oxide synthase 3 (NOS3) gene in primary cultures of endothelial cells from umbilical, fetal aorta, and femoral arteries of IUGR guinea pigs’ model [159].

### 4.5. MitoQ

In a hypoxic pregnancy of a rodent animal model, MitoQ was demonstrated to restore the placental levels of the activating transcription factor (ATF4), preventing the activation of mitochondrial oxidative stress and the unfolded protein response (UPR) signaling pathway [160]. Another interesting finding of this study was that MitoQ was demonstrated to increase placental volume, the fetal capillary surface area in the labyrinth zone, and the volume of maternal blood spaces in the placenta in this rat hypoxic pregnancy model [160]. The authors suggested that this mechanism may improve substrate supply to the fetus as a consequence of the increase in nitric oxide (NO) bioavailability. In fact, NO is essential to maintain both endothelial function and umbilical blood flow [5].

Despite the antioxidant compounds’ potential contribution to endothelial cell homeostasis and oxidative stress reduction, ROS acts as second messengers to the activation of several cellular signaling pathways, namely trophoblastic proliferation, which is crucial during the early stages of pregnancy [161]. A recent study reported that although MitoQ oral gavage administration decreased placental oxidative damage markers (MDA and 4-HNE) both during early and late gestation in a reduced uterine perfusion pressure (RUPP)-induced mouse preeclampsia model, at an early gestation stage, multiple negative outcomes were observed [161]. MitoQ-treated preeclampsia mice mothers presented fetal loss 2 days after MitoQ administration and a significant increase in systolic and diastolic blood pressure [161]. Moreover, fetal and placental weight were significantly lower in the treated group as well as in the labyrinth/spongiotrophoblast ratio and the density of placental blood sinuses [161]. This study suggests that further studies are demanded regarding antioxidant therapy in early pregnancy.

Overall, it seems that maternal antioxidant supplementation exerts potential benefits to mitigate, or even prevent, pregnancy-related disorders, mainly via three different ways including through direct action on maternal metabolism, by improving fetoplacental function, and through direct action on the offspring.

## 5. Future Perspectives

Given the role of ROS as signaling molecules, it is critical to diverge future research into understanding whether an optimal concentration of each supplement can be reached without compromising whole-body signaling and metabolic function and whether these can be defined throughout the whole stages of pregnancy. Thus, there is still a great window left to explore the effects of maternal antioxidant supplementation in the offspring exposed to an adverse intra-uterine environment. It remains imperative to explore strategies targeted towards mitochondria and oxidative stress, which may prevent metabolic dysregulation and pregnancy-associated disorders inducing metabolic dysregulation and oxidative stress both in the fetoplacental unit and in the offspring. Indeed, given the efficacy of new antioxidants, such as AntiOxBEN2 and AntiOxCIN4 in multiple oxidative stress- and mitochondrial dysfunction-related pathologies, it would be interesting to evaluate whether these antioxidants are able to target pregnancy-related disorders’ adverse effects on maternal and offspring’s health.

## 6. Conclusions

This review highlights the potential link between pregnancy-related disorders, such as GDM, MO, and IUGR, and the increased predisposition of chronic diseases in the offspring later in life. Pregnancy-related disorders appear to contribute to mitochondrial dysfunction and oxidative stress both in the fetoplacental unit and in the offspring’s organs (i.e., liver, pancreas, adipose tissue, and heart). Furthermore, this review presents compelling evidence supporting that maternal antioxidant supplementation, including resveratrol, curcumin, and NAC, may offer a promising strategy for mitigating metabolic dysfunction and oxidative stress induced by pregnancy-related conditions in the mothers, in the fetoplacental unit, and in the offspring. This area of research on maternal antioxidant administration warrants further investigation and attention as a potential means to prevent or mitigate the offspring’s metabolic disease programming to chronic diseases.

## Figures and Tables

**Figure 1 nutrients-15-04623-f001:**
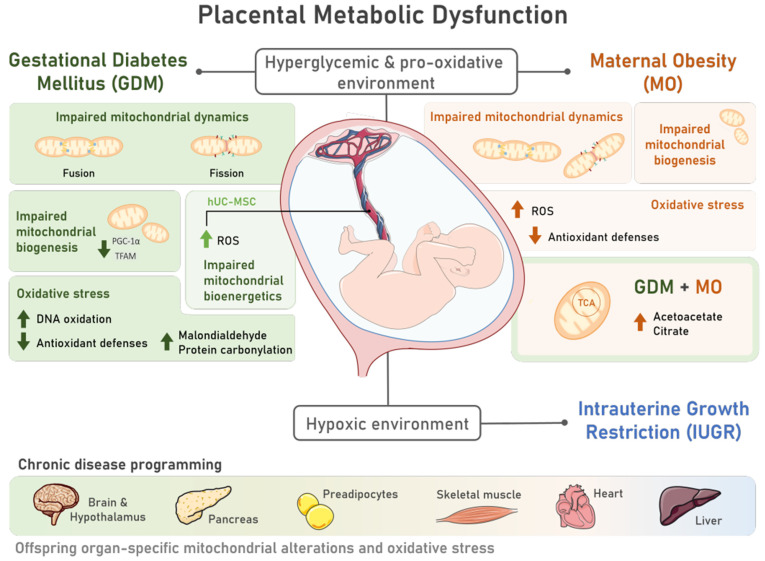
Placental metabolic dysfunction during pregnancies characterized by the presence of Gestational Diabetes Mellitus (GDM) and Maternal Obesity (MO) includes mitochondrial structural and functional alterations and oxidative stress, contributing to offspring chronic disease programming. The characteristic hyperglycemic and pro-oxidative intrauterine environment of pregnancies complicated by GDM and MO induces placental metabolic dysfunction via alterations in mitochondrial dynamics, with unbalanced fission and fusion events; mitochondrial biogenesis, via decreased peroxisome proliferator-activated receptor gamma coactivator 1-alpha (PGC-1α) and mitochondrial transcription factor A (TFAM) levels; and impaired mitochondrial bioenergetics, which was also observed in multipotent human umbilical cord mesenchymal stem cells (hUC-MSC) of GDM pregnancies. Increased reactive oxygen species (ROS) are also common to both placenta and hUC-MSC tissues in both pregnancy disorders. Placental oxidative stress is further marked by reduced antioxidant defenses and increased DNA oxidation, protein carbonylation, and malondialdehyde levels in GDM placentas. MO during a GDM pregnancy may further contribute to mitochondrial dysfunction, as increased levels of metabolites involved in the tricarboxylic acid (TCA) cycle and ketogenesis (citrate and acetoacetate, respectively) were detected. This compromised intrauterine environment can progress into a hypoxic state, increasing the risk of Intrauterine Growth Restriction (IUGR) development. GDM, MO, and IUGR have been associated with an increased risk of offspring chronic disease. Mitochondrial alterations and oxidative stress, which are intimately involved in chronic diseases, were observed in human and animal studies in different tissues of GDM and MO offspring, such as the brain and hypothalamus, pancreas, preadipocytes, skeletal muscle, heart, and liver, highlighting the role of metabolic pregnancy disorders in disease programming.

**Figure 2 nutrients-15-04623-f002:**
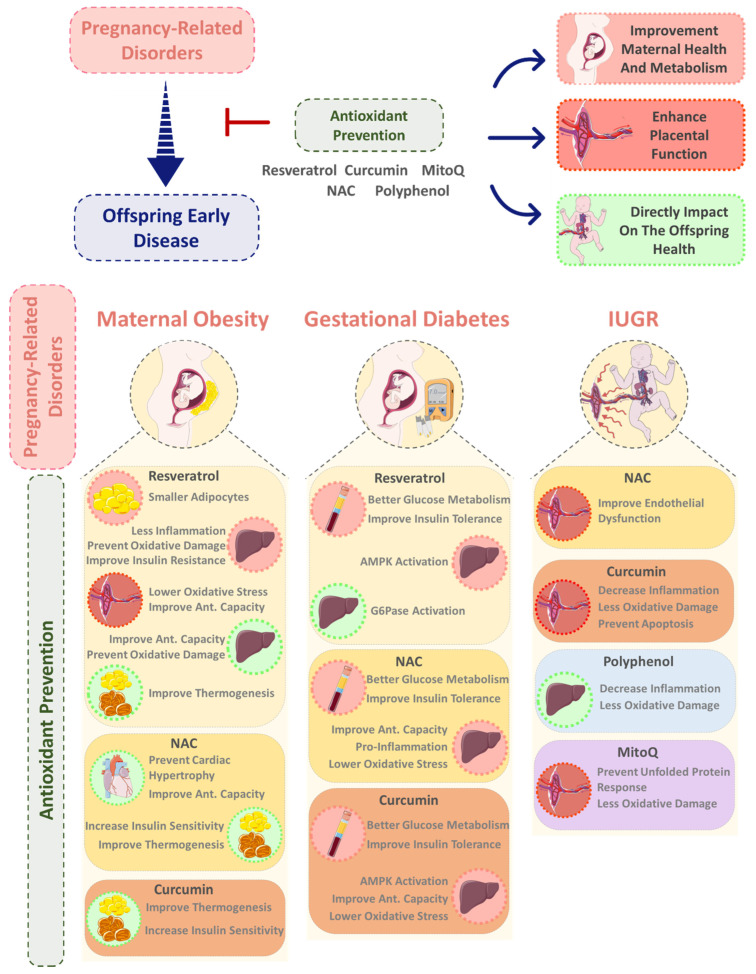
Maternal antioxidant supplementation/administration in preventing adverse intrauterine environments (Maternal Obesity (MO), Gestational Diabetes Mellitus (GDM), and Intrauterine Growth Restriction (IUGR))-induced deleterious developmental programming of the offspring. Antioxidants may act by improving maternal metabolism and health during pregnancy; by preventing placental dysfunction associated with adverse maternal conditions; or by directly preventing fetal abnormalities during development and developmental programming of disease. Multiple antioxidants (e.g., *N*-Acetylcysteine (NAC), polyphenols (i.e., resveratrol and curcumin), and MitoQ) were shown to act in the maternal liver, white adipose tissue (WAT), and blood parameters; placental function; and offspring liver, heart, WAT, and brown adipose tissue (BAT).

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
