# Peer review of "Nurturing through Nutrition: Exploring the Role of Antioxidants in Maternal Diet during Pregnancy to Mitigate Developmental Programming of Chronic Diseases"

_nutrients, 2023, doi:10.3390/nu15214623_

Round 1

Reviewer 1 Report

Comments and Suggestions for Authors

This manuscript presented a review about the role of antioxidants in maternal diet during pregnancy to mitigate fetal programming of chronic diseases, and include some evidence that mitochondrial dysfunction and oxidative stress are mechanisms by which GDM, MO, and IUGR can program the offspring to chronic diseases. The written English is good; nevertheless, the current manuscript still needs moderate revision because some questions are found.

1. The whole manuscript did not well match the title. For an instance, the introduction section, the foreword did not focus well on the topic and the logic is relatively chaotic. In fact, the logic of the whole manuscript is not good enough.

2. Some references are inaccurate; some results should be given with literature, eg. Line 369-371. Please check similar issues throughout the manuscript.

3. Figure 2 is too blurry.

   A review should be comprehensive, scientific, and logical, so it should be well organized. I hope the authors could give a full revision.

Author Response

This manuscript presented a review about the role of antioxidants in maternal diet during pregnancy to mitigate fetal programming of chronic diseases, and include some evidence that mitochondrial dysfunction and oxidative stress are mechanisms by which GDM, MO, and IUGR can program the offspring to chronic diseases. The written English is good; nevertheless, the current manuscript still needs moderate revision because some questions are found.

We genuinely appreciate your thoughtful review of our manuscript. Your constructive feedback regarding the need for moderate revisions is duly noted. We are committed to addressing the questions you've raised and further improving the manuscript. Your input has significantly contributed to enhancing the quality and clarity of our work, and we look forward to receiving your comments on this even more refined and comprehensive version in response to your valuable suggestions. Thank you for your insights!

  1. The whole manuscript did not well match the title. For an instance, the introduction section, the foreword did not focus well on the topic and the logic is relatively chaotic. In fact, the logic of the whole manuscript is not good enough.

We appreciate your time and comments that helped us improve our manuscript. Considering the reviewer’s comment, we significantly revised our manuscript. In the introduction section, we re-structured the text in lines 62-78. In section 2.2., we added additional information in lines 155, 167-170, and 232-245. Section 3 was completely re-structured: sub-sections were created to separate naturally occurring antioxidants (endogenous antioxidants – section 3.1.1 and dietary antioxidants – section 3.1.2) from synthetic antioxidants, which was sub-sectioned into “mitochondria-targeted antioxidants – section 3.2.1”. Moreover, to improve clarity, section 4 was re-organized. We sub-divided into 4 different sections, each dedicated to studies in different antioxidants (4.1 Resveratrol, 4.2 Curcumin, 4.3 NAC, 4.4 MitoQ) in the prevention of GDM-, MO-, and IUGR-induced impairment in the placenta and offspring’s organs. As a result, the manuscript is now well-organized, coherent, and logically structured to effectively convey its intended message in alignment with the title.

  1. Some references are inaccurate; some results should be given with literature, eg. Line 369-371. Please check similar issues throughout the manuscript.

Thank you for your valuable feedback regarding the accuracy of references and the need for additional contextualization of results. The issues were corrected, and similar issues were thoroughly checked. We greatly appreciate your attention to detail, as it reinforces the scholarly quality of our work!

  1. Figure 2 is too blurry.

Figure 2 was initially uploaded with a notably higher resolution compared to what was presented in the manuscript submitted for reviewers' evaluation. Recognizing the importance of maintaining this enhanced quality in the published version, we have taken proactive steps to address this issue. Consequently, we have now uploaded individual PDF versions of each figure within the system to ensure optimal resolution and clarity

A review should be comprehensive, scientific, and logical, so it should be well organized. I hope the authors could give a full revision.

Thank you for your valuable feedback and your expectation of a comprehensive, scientific, and logically organized review. We want to assure you that we have taken your comments seriously, and the issues you raised have been carefully addressed. The manuscript has undergone a thorough revision, including reorganization for improved logical flow and clarity. We believe that the changes made will significantly enhance the overall quality and readability of the paper. We appreciate your input, which has been instrumental in refining our work. Your expectations have been met with a dedicated commitment to delivering a more polished and scientifically sound manuscript.

Reviewer 2 Report

Comments and Suggestions for Authors

  1. The review "Nurturing Through Nutrition: Exploring the Role of Antioxidants in Maternal Diet during Pregnancy to Mitigate Fetal Programming of Chronic Diseases" discusses the potential role of antioxidant supplementation in mitigating the developmental programming of chronic diseases in offspring. While the topic is of great relevance given the rising global burden of non-communicable chronic diseases, there are several weaknesses in the manuscript that require attention before publication. The review should include a balanced discussion of potential risks or limitations associated with maternal antioxidant supplementation, as indiscriminate use of supplements during pregnancy can have adverse effects. More emphasis should be placed on summarizing and critically analyzing existing research studies that investigate the effects of maternal antioxidant supplementation during pregnancy. The introduction provides a clear overview of the research topic, outlining the connection between pregnancy-related conditions (GDM, MO, and IUGR) and the risk of chronic diseases in offspring. It also hints at the role of mitochondrial function and oxidative stress. However, it lacks specific background information to introduce the reader to these concepts. A brief explanation of the importance of mitochondrial function and oxidative stress in cellular processes and development could improve the introduction. In part2, the section contains complex scientific terminology and concepts that may be challenging for non-expert readers to grasp. Consider simplifying explanations and providing clearer definitions for key terms.

Author Response

We greatly appreciate the insightful feedback provided in the review of our manuscript titled "Nurturing Through Nutrition: Exploring the Role of Antioxidants in Maternal Diet during Pregnancy to Mitigate Fetal Programming of Chronic Diseases." We have taken your suggestions into careful consideration and incorporated them into the revised version of the manuscript. We truly value your constructive critique, which has been instrumental in improving the manuscript's quality and clarity. Your insights have greatly contributed to making our research more robust and reader-friendly, and we look forward to sharing this refined version with you for further feedback and evaluation. Thank you for your thoughtful and valuable suggestions.

Reviewer 3 Report

Comments and Suggestions for Authors

Manuscript „Nurturing Through Nutrition: Exploring the Role of Antioxidants in Maternal Diet during Pregnancy to Mitigate Fetal Programming of Chronic Diseases” presents the possible mechanisms of offspring’s chronic disease programming by maternal health and discuss possible beneficial effects of supplementation with selected antioxidant compounds.

Strengths: Extensive manuscript with 150 publications. Well written, 2 figures.

Weak points:

Generally, the text is written well and clearly. However, the presentation of both Figures(1 and 2) requires refinement. There is a reference in the text to Figure 1/2, and then an entire paragraph is devoted to the description of Figure 1/2. I think it should be combined so as not to repeat information. Furthermore, figure ½ themes are missing and the quality of figure 2 is very low.

Author Response

Manuscript “Nurturing Through Nutrition: Exploring the Role of Antioxidants in Maternal Diet during Pregnancy to Mitigate Fetal Programming of Chronic Diseases” presents the possible mechanisms of offspring’s chronic disease programming by maternal health and discuss possible beneficial effects of supplementation with selected antioxidant compounds.

Strengths: Extensive manuscript with 150 publications. Well written, 2 figures.

We sincerely appreciate your positive feedback on our manuscript, "Nurturing Through Nutrition: Exploring the Role of Antioxidants in Maternal Diet during Pregnancy to Mitigate Fetal Programming of Chronic Diseases." Your recognition of its strengths is encouraging, and we are pleased to know that you found it well-written and appreciated the inclusion of two figures.

Weak points:

Generally, the text is written well and clearly. However, the presentation of both Figures(1 and 2) requires refinement. There is a reference in the text to Figure 1/2, and then an entire paragraph is devoted to the description of Figure 1/2. I think it should be combined so as not to repeat information. Furthermore, figure 1/2 themes are missing and the quality of figure 2 is very low.

We have enhanced the quality of Figure 2. It's important to clarify that the paragraphs mentioned by the reviewer are integral parts of the manuscript and not figure legends. However, the figure legends were initially included in the document uploaded during the submission process but regrettably omitted in the version sent for reviewers' evaluation. To rectify this oversight, we have uploaded a revised manuscript that includes individual PDFs for each figure, complete with their respective legends.
